# Plant Based Diet and Its Effect on Cardiovascular Disease

**DOI:** 10.3390/ijerph20043337

**Published:** 2023-02-14

**Authors:** Salman Salehin, Peter Rasmussen, Steven Mai, Muhammad Mushtaq, Mayank Agarwal, Syed Mustajab Hasan, Shahran Salehin, Muhammad Raja, Syed Gilani, Wissam I. Khalife

**Affiliations:** 1Department of Internal Medicine, University of Texas Medical Branch, Galveston, TX 77555, USA; 2Department of Cardiology, University of Texas Medical Branch, Galveston, TX 77555, USA; 3School of Medicine, Holy Family Red Crescent Medical College Hospital, Dhaka 1000, Bangladesh

**Keywords:** plant based diet, vegetarian diet, nutrition, cardiovascular disease, preventive cardiology

## Abstract

Cardiovascular disease remains the leading cause of death globally and here in the United States. Diet has a major impact on the pathogenesis of atherosclerosis and subsequent cardiovascular morbidity and mortality. An unhealthy diet is the most significant potential behavioral and modifiable risk factor for ischemic heart disease. Despite these established facts, dietary interventions are far less frequent than pharmaceutical and procedural interventions in the management of cardiovascular disease. The beneficial effects of a plant-based diet on cardiovascular morbidity and mortality have been demonstrated in a number of recent clinical studies. The significant findings of each study are discussed in this review article, highlighting the role of a healthy plant-based diet in improving cardiovascular outcomes. From a clinician’s standpoint, the knowledge and understanding of the facts and data points from these recent clinical studies would ensure more effective patient counseling on the substantial benefits of dietary interventions.

## 1. Introduction

According to the World Health Organization, Cardiovascular Disease (CVD) remains the paramount cause of death globally, leading to the loss of an estimated 17.9 million lives in 2019 [1]. According to the Centers for Disease Control and Prevention (CDC), heart disease is also the major cause of death in the United States, costing the lives of 697,000 people in 2020.

Atherosclerosis and its subsequent cardiovascular morbidity and mortality are primarily affected by diet [2]. The most important potential behavioral risk factor for ischemic heart disease (IHD) is an unhealthy diet, which has a high content of sodium, trans fats and processed meats but low amounts of vegetables, fruits, nuts and seeds [1,2]. Despite the aforementioned established facts, dietary interventions are much less common than pharmacological and procedural interventions in the management of cardiovascular disease [2].

Plant-based diets can lower all-cause mortality and lower the risk of ischemic heart disease with reduced IHD-related mortality [3,4]. It can also optimize blood pressure [5,6,7], glycemic [8,9] and lipid control [8,9,10,11], and thus reduce the need for medications [8,11].

According to the 2019 ACC/AHA (American College of Cardiology/American Heart Association) guidelines on nutrition, a diet emphasizing intake of vegetables, fruits, legumes, nuts, whole grains and fish is recommended to decrease Atherosclerotic Cardiovascular Disease (ASCVD) risk (Class I recommendation). The replacement of saturated fat with dietary monounsaturated and polyunsaturated fats coupled with a reduction in amounts of sodium, cholesterol, processed meats, refined carbohydrates and sweetened beverages can be beneficial in reducing ASCVD risk (Class IIa recommendation).

Several recent clinical studies have exhibited the beneficial effects of a plant-based diet on cardiovascular morbidity and mortality. Articles were identified by searching keywords in PubMed such as plant based, cardiovascular outcomes and vegan diet [12]. In this review article, we discuss the significant findings of each study, emphasizing the role of a healthy plant-based diet in improving cardiovascular outcomes. A greater understanding of these studies could help clinicians guide diet recommendations and improve cardiovascular outcomes for their patients.

## 2. Trials

### 2.1. Adventist Health Study 2 and Subsequent Investigations

#### 2.1.1. Background

In recent years, as more data has emerged regarding the health threats associated with processed meats and other animal-based foods, many Americans have chosen to adopt either a vegetarian or vegan lifestyle. According to a 2018 Gallup poll, 5% and 3% of Americans consider themselves vegetarians and vegans, respectively. The driving force behind this decision revolves around the public’s desire to prolong longevity and avoid the ill effects of common illnesses that likely have a dietary component, such as cardiovascular disease, diabetes and some cancers. Prior studies have attempted to determine which dietary choice is associated with the best health outcomes. For example, the first Adventist Health Study assessed approximately 34,000 Californian Seventh-day Adventists who practice vegetarianism and found that the vegetarian diet reduced all-cause mortality and increased life expectancy [13]. On the other hand, the European Prospective Investigation into Cancer and Nutrition-Oxford (EPIC-Oxford) cohort study did not find a difference in all-cause mortality between vegetarians and non-vegetarians [14]. As demonstrated by the literature, the impact of a vegetarian or vegan diet on health outcomes is not fully understood. To address this knowledge gap, the Adventist Health Study 2 (AHS-2) was conducted through Loma Linda University in California [15]. AHS-2 is an ongoing prospective observational cohort study that seeks to determine the link between mortality and dietary practices among a large American and Canadian cohort that includes many vegetarians and vegans [3].

#### 2.1.2. Population

Individuals for the AHS-2 study were recruited from Seventh-day Adventist churches throughout all of Canada and America between 2002 and 2007. Since the inception of AHS-2, a total of 96,469 individuals were recruited via lifestyle questionnaires distributed among the churches. After applying exclusions, a final analytic sample of 73,308 remained. The following exclusion criteria were applied: missing data for questionnaire, non-US residents, past history of specific cancer or cardiovascular disease and estimated energy intake of less than 500 kcal/day or more than 4500 kcal/day. The mean age of the study population is 60.2, 65.1% are female, and the ages range from 30 to 112. A total of 65.3% are white and 26.9% are African American. Only 1.2% and 1.0% are male and female smokers, respectively. Individuals who consumed less than 500 or greater than 4500 kcal per day as well as those with a history of cancer or cardiovascular disease were excluded.

Individuals were categorized into five dietary practices including non-vegetarian, semi-vegetarian, pesco-vegetarian (avoids meat/animal flesh but consumes fish), lacto-ovo-vegetarian (consumes dairy and egg products), and vegan (avoids any animal product). Other basic demographic data including sex, race, geographic region, educational level, smoking and alcohol use as well as exercise frequency were included in the study as well. Cox proportional hazards regression was used to assess the data. A total of 5548 (7.6%) were vegans, 21,177 (28.9%) were lacto-ovo-vegetarians, 7194 (9.8%) were pesco-vegetarians, 4031 (5.5%) were semi-vegetarians, and 35,359 (48.2%) were non vegetarians.

#### 2.1.3. Outcomes to Date

The overall mortality rate was 6.05 deaths per 1000 person-years (95% CI, 5.82–6.29). Vegetarians had a hazard ratio of 0.88 (95% CI, 0.8–0.97) for all-cause mortality when compared to non-vegetarians. Similarly, the adjusted hazard ratio (HR) for all-cause mortality among vegans was 0.85 (95% CI, 0.73–1.01), 0.91 among lacto-ovo-vegetarians (95% CI, 0.82–1.00), 0.81 among pesco-vegetarians (95% CI, 0.69–0.94), and 0.92 among semi-vegetarians (95% CI, 0.75–1.13) when compared to non-vegetarians. Pesco-vegetarians and vegan men had a lower risk of mortality from ischemic heart disease (IHD) compared to that of non-vegetarians (HR 0.65 95% CI, 0.43–0.97; HR 0.45, 95% CI, 0.21–0.94, respectively). Vegetarian men had a statistically significant HR of 0.82 (95% CI, 0.72–0.94) for all-cause mortality, while vegetarian women had a statistically insignificant HR of 0.93 (95% CI, 0.82–1.05). Similarly, vegetarian men had a statistically significant HR of 0.71 (95% CI, 0.57–0.90) for cardiovascular mortality, while vegetarian women had a statistically insignificant HR of 0.99 (95% CI, 0.83–1.18).

In addition to heart disease and all-cause mortality benefits, rates of hypertension, diabetes, cancer and osteoporosis were also lower in some vegetarian groups compared to non-vegetarians [16]. In a logistic regression analysis adjusted for age, sex and exercise frequency, the odds ratio (OR) of having hypertension was 0.37 (95% CI, 0.19–0.74) for vegans when compared to non-vegetarians [5]. Among vegans and non-vegetarians, 2.9% and 7.6% reported having type 2 diabetes mellitus (DM2), respectively. In a similar logistic regression model, the OR for DM2 prevalence was 0.51 (95% CI, 0.40–0.66) in vegans when compared to non-vegetarians. Moreover, at a 2-year follow up, 0.54% of vegans compared to 2.12% of non-vegetarians reported developing DM2 (OR 0.38, 95% CI 0.24–0.62) [17].

Despite multiple dietary practices that have demonstrated improved health outcomes, the specific foods responsible for these outcomes are still partially unknown. Tharrey et al. addressed a portion of this broad question by analyzing protein sources and their relation to health outcomes [18]. Using the 2276 cardiovascular deaths that occurred during the mean follow up of 9.4 years since the study’s inception in 2002, researchers found that the HR for CVD was 1.61 (98.75% CI, 1.12–2.32) for participants who primarily obtained their protein from meat compared to 0.60 (98.75% CI, 0.42–0.86) for those who obtained their protein from nuts and seeds. The authors concluded that a diet sourcing protein primarily from plants reduces the risk of cardiovascular mortality. This finding may be related to the higher concentration of glutamic acid in plant proteins compared to animal proteins, which has been linked to lower blood pressure [19]. Additionally, nuts also have a high L-arginine content, which is converted to nitric oxide and may play a positive role in cardiovascular health [20].

#### 2.1.4. Discussion

The current data from the AHS-2 study suggests that adopting a vegetarian or vegan diet may result in all-cause and cardiovascular mortality reductions in addition to prolonging longevity. Although the exact mechanism by which these diets improve health outcomes is unclear, the literature suggests that it may be related to a more favorable cholesterol profile and less-frequent consumption of processed foods as well as pro-inflammatory substances found in animal-based products [21]. An interesting insight from AHS-2 includes a notable stronger positive effect of a vegetarian or vegan diet on outcomes in men when compared to women. It is unclear why men tend to have a greater improvement in all-cause mortality and other key endpoints including IHD mortality when compared to women. Further investigation into these sex-specific associations is warranted. Overall, the results of this prospective observational study suggest that moving away from a traditional non vegetarian diet may be beneficial. However, the exact diet practice or specific food choices for optimal health outcomes remains unclear.

#### 2.1.5. Limitations

Although there are many strengths of the AHS-2 study, several limitations should be noted. First, a short follow-up period may hide the true effects of diet choice on long term mortality outcomes, especially if these lifestyle choices modify risk through complex time-dependent mechanisms. Additionally, the constellation of foods that each vegetarian consumes can widely vary. As noted by the authors, further analysis into specific possible culprit foods such as processed dairy products or foods high in saturated fats should be studied. Lastly, participant reporting bias of their true diets may be skewed towards reporting more favorable diet profiles.

### 2.2. European Prospective Investigation into Cancer and Nutrition—Oxford (EPIC-Oxford) Study

#### 2.2.1. Background

The EPIC-Oxford study was designed to examine an association between ischemic heart disease (IHD) or stroke and vegetarianism [22]. This prospective cohort study included participants without a history of IHD, stroke or other CV disease. Of the 48,188 British participants recruited between the years of 1993 and 2001, three diet groups were created. The first group was labeled meat eaters. The second group was labeled fish eaters, which were those who consumed fish but avoided meat, and the third group were vegetarians and vegans. A total of 24,428 were meat eaters, 7506 were fish eaters, and 16,254 were vegetarians. The mean age of the meat eaters, fish eaters, and vegetarians was 49, 42.1, and 39.4, respectively. Among meat eaters, fish eaters, and vegetarians, 75.7%, 82.4%, and 75.3% were women, respectively. Mean BMI among meat eaters, fish eaters, and vegetarians was 24.1, 23.1, and 23, respectively. The primary outcome measure was the incidence of IHD or stroke over 18 years of follow up.

Participants were recruited in a similar manner to that of the Adventist study in that both employed a questionnaire to enroll patients. One notable difference is that the Adventist study recruited patients mostly from churches, whereas the EPIC-Oxford study recruited participants from physician offices, vegetarian societies, and cohorts from the prior Oxford Vegetarian study. The EPIC-Oxford study was similar to the Adventist study in the sense that it also accounted for demographic information including education level, smoking status, frequency of exercise, and socioeconomic status. Cox proportions hazard regression models were also used in both the Adventist and EPIC-Oxford studies.

#### 2.2.2. Outcomes

Similar to the findings of the Adventist trial, non-meat eaters were more highly educated, less likely to smoke or consume alcohol and more physically active. Moreover, vegetarians had lower cholesterol levels when compared to meat and fish eaters. Regarding cardiac and stroke outcomes, there were a total of 2820 cases of IHD and 1072 cases of strokes. Vegetarians were less likely to develop IHD or have a stroke when compared to meat eaters (HR 0.78, 95% CI 0.7–0.87). On the other hand, the EPIC-Oxford study is unique in that it demonstrated a higher rate of total stroke in vegetarians when compared to meat eaters (HR 1.2, 95% CI 1.02–1.4). Likewise, the EPIC-Oxford study did not find a significant difference between the diet groups regarding the risk of acute myocardial infarction or ischemic stroke [23].

#### 2.2.3. Discussion

The cardiovascular outcomes of the EPIC-Oxford study were similar to the findings of previous large prospective studies, including the Adventist Mortality Study, Health Food Shoppers Study, Adventist Health Study, Heidelberg Study and the Oxford Vegetarian Study. A meta-analysis of these five previous studies found that vegetarians had a 24% lower ratio of death from IHD when compared to non-vegetarians [24]. On the other hand, prior studies had not shown a significant difference in stroke mortality based on diet choices [25]. Similar to the Adventist study, the EPIC-Oxford authors hypothesized that the improvements in cardiovascular and stroke outcomes may be related to lower concentrations of low-density lipoprotein cholesterol (LDL-C) associated with vegetarian diets [26]. Moreover, an observational study by Sun et al. published in 2019 found that participants with lower LDL-C levels were more likely to sustain a hemorrhagic stroke [27]. This finding echoes the finding of the EPIC-Oxford study and may point to a possible underlying relationship. Similarly, Kinjo et al. found that individuals in Japan who had a low intake of animal products were at a higher risk for mortality from hemorrhagic strokes [28]. This study, in combination with the EPIC-Oxford and Sun et al. study, suggests that animal products or LDL-C may have a protective role against hemorrhagic stroke. The EPIC-Oxford authors also noted that vegetarians had lower blood pressures compared to meat eaters, making hypertension an unlikely explanation for this finding. Albeit interesting, misclassification of stroke-type in the patient data may have biased these results.

#### 2.2.4. Limitations

Similar to the Adventist study, the EPIC-Oxford study utilized a large sample size with a long follow-up period, but the results may have been biased by the subjective reporting of dietary practices. Furthermore, generalizing the findings of the EPIC-Oxford study to other countries or races may be difficult, for the study was predominately performed on Caucasian European individuals.

### 2.3. Plant-Based Diets Are Associated with a Lower Risk of Incident Cardiovascular Disease, Cardiovascular Disease Mortality, and All-Cause Mortality in a General Population of Middle-Aged Adults: Data from the ARIC Study

This study by Kim et al. was a *prospective cohort study* that analyzed patient data from a community-based cohort in the ARIC (Atherosclerosis Risk in Communities) study to assess if overall plant-based diets are associated with a lower risk of incident cardiovascular disease, cardiovascular disease mortality and all-cause mortality in a general US population [29]. Additionally, it assessed that the association differed by adherence to healthful and less healthful plant-based diets. A total of 15,792 patients were followed with follow-up visits from 1990 to 1992, 1993 to 1995, 1996 to 1998, 2011 to 2013, and 2016 to 2017.

#### 2.3.1. Patient Population

Patients were from the ARIC study. These patients were from four US communities, including Washington county, Maryland; Forsyth County, North Carolina; Minneapolis, Minnesota; and Jackson, Massachusetts. The patients were 45–64 years old and selected between 1987 and 1989. Patients were assessed on adherence of diet utilizing four index scores that included Plant based diet index (PDI), healthy plant-based diet index (hPDI), less healthy plant-based diet (uPDI) and pro-vegetarian diet index groups, then separated into quintiles within each score. They were further separated into three models adjusting for various characteristics. The first model adjusted for total energy intake, age, sex and race-center. The second model adjusted for education, cigarette smoking, physical activity, alcohol intake and margarine intake. The third model adjusted for total cholesterol, lipid-lowering medication use, hypertension, diabetes, kidney function and BMI.

#### 2.3.2. Primary Endpoint and Results

Primary endpoints included cardiovascular disease incidence (CDI), cardiovascular disease mortality (CVM) and all-cause mortality, which were compared between the accumulated quintiles, between the various model adjusted quintiles and amongst the individual components and food groups of the indexes.

#### 2.3.3. Comparisons amongst Quintiles of the Indexes

At the 25-year follow up, the CVM and all-cause mortality were significantly lower at higher quintiles of PDI, with *p* < 0.001 and *p* < 0.001, respectively. There was a significant difference amongst hPDI quintiles when comparing CVM (*p* = 0.01) and all-cause mortality (*p* = 0.01) too. Finally, there was a significant difference in CVM (*p* < 0.001) and all-cause mortality (*p* < 0.001) amongst pro-vegetarian quintiles. Meanwhile, there were no significant differences across quintiles of uPDI for CVM (*p* = 0.13) or all-cause mortality (*p* = 0.10). CDI was significantly different across quintiles of PDI (*p* < 0.001) and pro-vegetarian diets (*p* < 0.001), but it was insignificant in hPDI (*p* = 0.11) and uPDI (*p* = 0.98) patients.

#### 2.3.4. Model-Adjusted Comparisons amongst Index Quintiles

The strongest associations were observed for PDI and pro-vegetarian diet indexes, which had a significant difference in primary outcomes in all three models. A significant difference was noted amongst quintiles of model-1-adjusted PDI patients for CDI, CVM and all-cause mortality, all with *p* < 0.001. Additionally, model 2- and 3-adjusted PDI was significant with *p* < 0.001 for CDI, CVM and all-cause mortality. Model-1-adjusted pro-vegetarian had a significant *p* < 0.001 for CDI, CVM and all-cause mortality. Additionally, model 2- and 3-adjusted pro-vegetarian patients were significant for CDI, CVM and all-cause mortality, with a *p* < 0.001. Those in the highest quintiles of PDI and pro-vegetarian had a 16% and 16% lower risk of CDI, a 32% and 31% lower risk in CVM, and a 25% and 18% lower risk in all-cause mortality.

The hPDI patients in model 2 had a 19% lower risk of CVM (*p* = 0.01) and 11% lower risk of all-cause mortality (*p* = 0.01) when comparing higher to lower quintiles. Model 3 hPDI patients additionally had a significant difference in CVM (*p* = 0.03) and all-cause mortality (*p* = 0.03). Despite the significance in CVM, models 2 and 3 had no significant association when comparing CDI between quintiles (*p* = 0.56).

Lastly, the uPDI group had no significant associations in outcomes, regardless of the adjusted models. The *p*-values were *p* = 0.53 for CDI, *p* = 75 for CVM and 0.95 for all-cause mortality in model 1 adjusted. The *p*-values were *p* = 0.48 for CDI, *p* = 0.94 in CVM and *p* = 0.67 for all-cause mortality in models 2 and 3 adjusted.

#### 2.3.5. Discussion

Ultimately, the study by Kim et al. supports the hypothesis that diets with higher adherence to plant foods are associated with improved cardiac morbidity and mortality in the general population. This remains the case regardless of adjustments for sex, age and socioeconomic status amongst other factors. The study further supports this when breaking down components of the indexes and showing that while there is a difference in the risk between quintiles of plant versus meat eaters, there is no significant difference between health vs. unhealthy plant-based food. The population utilized by Kim et al. allows for greater generalizability of the results compared to prior studies.

#### 2.3.6. Limitations

First, dietary intakes were self-reported, making them subject to error. Second, the sample-based scoring method used makes the study prone to errors and third factors. Finally, the diets utilized in this study may not accurately represent a modern diet, as the dietary intakes of patients in the ARIC study were measured several decades ago.

### 2.4. BROAD Study

The BROAD trial was a prospective, two-arm, parallel, superiority study published in 2017 by Wright et al. [30]. This study ran from August 2014 to 2015 and was extended to 2017; it assessed the effectiveness of whole food plant-based diets (WFPB) compared to a control group. A total of 65 patients were recruited and randomized in a 1:1 ratio to WFPB or control. End points were assessed at 6 and 12 months.

#### 2.4.1. Patient Population

Candidates were patients from a general practice in Gisborne, New Zealand and were 35 to 70 years old with BMI ≥ 30 kg/m^2^ and with a diagnosis of either type 2 diabetes, ischemic heart disease or cardiovascular risk factors of hypertension or hypercholesterolemia. The study excluded participants with life-threatening comorbidities, thyroid disease, coronary artery bypass grafting within 6 weeks, myocardial infarction within 1 month, >50% stenosis of the left main coronary artery, unresponsive congestive heart failure, malignant uncontrolled arrhythmias, homozygous hypercholesterolemia, severe mental health disorders, current alcohol or drug misuse, currently smoking, currently pregnant or breastfeeding women, prior bariatric surgery and other conditions that directly impact weight. A total of 693 candidates were invited via letter which involved EMR screening consent. After EMR screening, 116 candidates were interviewed and, ultimately, 65 participants were randomized.

#### 2.4.2. Primary Endpoints and Results

The primary endpoint focus was BMI and cholesterol. BMI and weight were statistically significant in reduction for both WFPB diet and between-group (*p* < 0.0001). The WFPB group had a mean BMI reduction of 4.4 kg m^−2^ (95% CI 3.7–5.1, *p* < 0.0001) and 4.2 kg m^−2^ (95% CI 3.4–5, *p* < 0.0001) at 6 and 12 months, respectively. The WFPB group had a 12.1 lbs (95% CI 9.9–14.3, *p* < 0.0001) and 11.5 lbs (95% CI 9–14, *p* < 0.0001) reduction at 6 and 12 months. Meanwhile, the control group had no significant difference at 3 (*p* = 0.2) or 6 months (*p* = 0.18).

Cholesterol reduction was statistically significant at all time periods: 0.95 mmol/L (95% CI 0.51–1.39, *p* < 0.001) at 3 months, 0.71 mmol/L (95% CI 0.28 to 1.14, *p* < 0.01) at 6 months, and 0.55 mmol/L (95% CI 0.01–1.09, *p* = 0.05) at 12 months. The control group had a significant decrease at Month 3 of 0.28 mmol/L (*p* = 0.03) but was not significant at 6 months (*p* = 0.15).

#### 2.4.3. Secondary Endpoint and Results

Secondary endpoints included changes in medication usage, quality of life, cardiovascular risk factors (CVDRA), cardiovascular events, progression to surgery, and transfer to a higher level of care. Additionally, at 6 months, personality inventory for factors associated with adherence were collected.

CVDRA in the intervention group decreased by 0.4% from baseline (*p* = 0.02) and the between-group was 0.6% (*p* = 0.02).Specifically, hemoglobin A1c (HgA1C) reduced by 5 at 6 (*p* < 0.001) and 12 (*p* < 0.0001) months. No changes were seen in the control group.There was no transfer to higher-level care or acute admission for any groups during the first 12 month of research.

#### 2.4.4. Conclusions

In conclusion, a WFPB diet will lead to a significant improvement in BMI, cholesterol and HgA1C without energy intake limitations or exercise requirements. Wright et al.’s trial resulted in greater weight loss in 6 and 12 months compared to other trials.

#### 2.4.5. Limitations

The limitations of this study include the explanation of the WFPB diet to all participants during the informed consent process that may have led to an increase in control group dietary indiscretions. In contrast, there was also a risk that the intervention group may not be perfectly adherent. Additionally, the inclusion of patients without necessarily elevated HgA1C or cholesterol may have dampened the effect size.

### 2.5. EVADE CAD Trial

Inflammation is associated with atherosclerotic progression and adverse cardiovascular events. Thus, unsurprisingly, major adverse cardiovascular events can be reduced by targeted anti-inflammatory therapies and decreased hsCRP (high sensitivity C-reactive protein). This point was again made evident by the recent CANTOS study, where the use of monoclonal antibody to interleukin-1β in patients with elevated hsCRP resulted in a significant reduction of major adverse cardiovascular events.

The EVADE CAD trial was a randomized control trial published in November 2018, which compared a plant-based vegan diet with the American Heart Association (AHA)-recommended diet at reducing inflammation and inflammatory markers such as hsCRP [21]. A total of 100 patients were recruited from a single-center in the US between March 2014 and February 2017 and randomized in a 1:1 ratio to either a vegan diet or the AHA-recommended diet. Endpoints were assessed at 8 weeks.

#### 2.5.1. Patient Population

The participants were of at least 18 years of age with a history of coronary artery disease (CAD) as per angiography (≥50% arterial lesion).

#### 2.5.2. Primary Endpoints and Results

The primary endpoint was concentration of hsCRP. After adjustment for baseline concentrations, there was a significant 32% lower concentration of hsCRP with the vegan diet when compared to the AHA diet [adjusted β estimate 0.67 (95% confidence interval 0.47–0.94), *p* = 0.02]. The result corroborates the anti-inflammatory effects of dietary fibers, although the exact mechanism remains unclear. One possibility is the restoration of gut microbiota with increased fibers which may improve the inflammatory profile.

(3.2% vs. 2.8%; hazard ratio [HR] 1.14, 95% confidence interval [CI] 0.80–1.62, *p* for noninferiority = 0.06.)

#### 2.5.3. Secondary Endpoints and Results

There were multiple secondary endpoints for this study, including but not limited to body mass index, glycemic markers, lipid panels and quality of life. No significant difference was observed between the AHA and vegan diet with respect to the aforementioned secondary endpoints. The vegan diet did bring about a 12% reduction in LDL cholesterol compared to the AHA diet after adjustment for baseline concentration. A composite of all-cause mortality, stroke or transient ischemic attack (TIA), myocardial infarction and repeat coronary revascularization was utilized to define major adverse cardiovascular and cerebrovascular events. Two participants in the AHA group had probable TIA, but no other major adverse cardiovascular or cerebrovascular events were noted.

#### 2.5.4. Conclusions

In conclusion, a vegan diet brought about a significant reduction in systemic inflammation in patients with CAD—as evidenced by the reduction in hsCRP, which is a marker of adverse cardiovascular outcomes—while the AHA diet did not. On the other hand, a vegan diet did not result in better glycemic control, more weight loss or lipid profile improvements when compared to the AHA diet. With these results in mind, it is important to understand how each diet impacts certain components of an individual’s overall health. Studies such as EVADE CAD can help providers recommend a diet that is tailored to each patient’s specific needs.

#### 2.5.5. Limitations

First, the generalizability may not extend outside of this trial’s patient population. Second, the power of the study was not sufficient to assess for significant differences in major adverse cardiovascular events. Third, the data was obtained by a head-to-head comparison of only two specific diets, namely the vegan and AHA-recommended diet. Fourth, underreporting of intake on the food records may have occurred on the part of the participants.

## 3. Discussion and Conclusions

Overall, the current literature supports the claim that vegetarian or vegan diets lead to more favorable health outcomes when compared to a traditional diet. The aforementioned studies have demonstrated that eliminating meat and increasing consumption of plant-based foods may prove beneficial for overall health. To summarize, the AHS-2 and ARIC studies demonstrated that plant-based diets are associated with improvements in cardiovascular mortality when compared to a standard diet. The EPIC-Oxford study found that vegetarianism is associated with lower rates of ischemic heart disease when compared to individuals who consume a regular diet that includes meat. While the BROAD study illustrated that a plant-based diet is associated with improvements in BMI, cholesterol and HgA1C, the EVADE CAD trial did not find a difference in these metrics when comparing vegans to those who follow the AHA diet. A summary of studies investigating the association between plant-based diets and cardiovascular disease has been presented in Table 1.

Based on these studies, a vegetarian or vegan diet has been associated with improvements in cardiovascular and all-cause mortality. The exact mechanism by which this occurs is not fully understood; however, several theories exist that may offer a reasonable explanation (Figure 1). In recent years, it has come to light that processed sugars, refined starches and high concentrations of saturated fat can induce low-grade systemic inflammation [31]. Similar to the EVADE CAD trial, a recent observational study published in January 2022 demonstrated that more frequent consumption of meat was associated with both a higher CRP and total white blood cell count [32]. This observation may be secondary to the higher concentration of meat-based heme-iron, which is known to induce inflammation when compared to the iron found in plants [33]. Chronic circulation of pro inflammatory mediators over time may predispose individuals to coronary atherosclerotic disease and subsequently, poor cardiac outcomes. Diets high in processed meats, dairy and other animal products have been shown to contribute to the circulation of pro inflammatory markers that promote endothelial dysfunction as well. For example, the health professionals follow-up study demonstrated that the individuals following a traditional western diet had higher circulating levels of CRP and tissue-type plasminogen-activating antigen when compared to those following a diet higher in vegetables, fruits and whole grains [34]. Similarly, in the Nurses’ health Study 1, women following a western diet had higher circulating levels of CRP and soluble vascular cell adhesion molecule 1 (sVCAM-1) when compared to those following a diet higher in vegetables, fruits and legumes [35]. Chronic exposure to higher levels of circulating CRP and other inflammatory mediators can promote plaque formation and subsequently lead to worsening cardiac outcomes. Furthermore, endothelial dysfunction promotes atherosclerosis as a result of platelet aggregation, increased endothelial permeability, cytokine production and leucocyte adhesion [36].

While many of the studies shed a positive light on plant-based and vegetarian diets, the overall literature remains mixed on the debate between plant-based and traditional diets. Some studies have not found an association between animal products and negative health outcomes. For example, a meta-analysis by Giosue et al. found that consuming up to 200 g of dairy per day was not associated with increased CVD risk [37]. Moreover, a large prospective cohort study by Budhathoki et al. demonstrated that animal protein consumption was not associated with cardiovascular, cerebrovascular or cancer mortality [38]. One notable limitation of all the included trials in our review is that the association of plant-based diets with developing CVD was not studied. Future investigations comparing diet choice with the development of CVD are warranted.

Another theory is that adopting a vegetarian or vegan diet may decrease the severity of conditions known to worsen cardiac outcomes, such as hypertension and diabetes, which in turn would lead to improvements in cardiac outcomes. Moreover, the lower concentrations of saturated fats and plasma levels of LDL-C associated with vegetarian or vegan diets may contribute to the improved longevity and mortality seen with these diet choices, as shown by the Adventist and EPIC-Oxford studies. Lastly, the importance of the gut microbiome’s relationship with health improvement has gained recognition in recent years. A vegan diet has been shown to promote the growth of beneficial bacterial flora, resulting in reduction in intestinal inflammation and improvements in nutrient absorption [39].

As the population continues to become more health conscious, the number of individuals following either vegetarian or vegan diets may increase. Emerging data highlighting new definitions of a healthy diet and what specific foods comprise that diet will ultimately guide formal dietary recommendations by major health organizations in the coming years. The current trend to pursue healthier choices may prolong longevity and improve cardiovascular outcomes over the next several decades.

## Figures and Tables

**Figure 1 ijerph-20-03337-f001:**
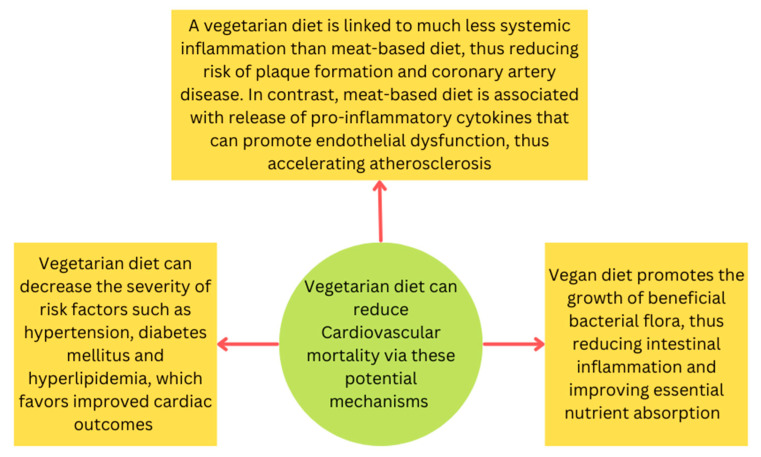
Potential mechanisms that explain favorable cardiovascular outcomes associated with a vegetarian or vegan diet.

**Table 1 ijerph-20-03337-t001:** Summary of Studies Investigating the Association Between Plant-Based Diets and Cardiovascular Disease.

Author	Study Design	Country	Year Published	Participants (n)	Age Range	Primary Results
Kim et al. [29]	Retrospective Cohort	United States	2019	15,792	45–64	CVM and all-cause mortality were lower at higher quintiles of PDI (*p* < 0.001, *p* < 0.001), hPDI (*p* = 0.01, *p* = 0.01), and pro-vegetarian diets (*p* < 0.001, *p* < 0.001) after 25 years of follow up. There were no significant differences in CVM and all cause quintiles of uPDI (*p* = 0.13, *p* = 0.10).
Wright et al. [30]	Randomized Control Trial	New Zealand	2017	65	35–70	Mean BMI reduction was greater in the WFPB group compared to those who received normal care (4.4 vs 0.4 kg/m^2^, *p* < 0.0001).
Orlich et al. [3]	prospective Cohort	United States and Canada	2013	73,308	30–112	Vegans had an adjusted HR Of 0.85 for all—cause mortality when compared to non-vegetarians (95% Cl, 0.73—1.01). Diet choice had a more significant impact on outcomes when compared to women.
Orlich et al. [16]	Prospective Cohort	United States and Canada	2014	73,308	30–112	The OR of having hypertension was 0.37 (95% CI, 0.19–0.74) for vegans when compared to non-vegetarians. The OR for DM2 prevalence was 0.51 (95% CI, 0.40–0.66) in vegans when compared to non-vegetarians. A composite of vegetarians and vegans had a lower risk of developing any type of cancer when compared to non-vegetarians (HR 0.92, 95% Cl 0.85–0.99)
Tharrey et al. [18]	Prospective Cohort	United States and Canada	2018	81,337	30–112	The HR for cardiovascular mortality was 1.61 (*p* < 0.001) for those who consume meat versus 0.60 (*p* < 0.001) for those who consume nuts and seeds as their protein source.
Tong et al. [22]	prospective Cohort	United Kingdom	2019	48,188	35–59	Vegetarians and pescatarians had 13% lower rates of ischemic heart disease compared to those who consume meat (*p* < 0.001). On the other hand, vegetarians had a 20% higher rate of total strokes, and particularly hemorrhagic strokes, when compared to meat eaters (HR 1.20, 95% CI, 1.02–1.40).
Key et al. [14]	Prospective Cohort	United Kingdom	2009	48,188	35–59	Meat eaters obtained 10.4% of their energy from saturated fats compared to 6.9% in vegans.
Shah et al. [21]	Randomized Control	United States	2018	100	53-68	Following a vegan diet resulted in a 32% lower concentration of hsCRP compared to following the AHA diet (*p* = 0.02). Participants following a vegan diet also had a 12% reduction in their LDL-C compared to those following the AHA diet.

Cardiovascular Mortality (CVM), Plant Based Diet Index (PDI), Healthy PDI (hPDI), Body Mass Index (BMI), Whole Food Plant Based (WFPB), Hazard Ratio (HR), Odds Ratio (OR), Diabetes Mellitus (DM), Confidence Intervel (CI), High Sensitivity C Reactive Protein (hsCRP), American Heart Association (AHA), Low Density Lipoprotein Cholesterol (LDL-C).

## Data Availability

Not applicable.

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
