# Peer review of "Plant Based Diet and Its Effect on Cardiovascular Disease"

_ijerph, 2023, doi:10.3390/ijerph20043337_

Round 1

Reviewer 1 Report

Dear authors, 

Hi, 

I suggest the following to improve the manuscript: 

1. In the figure, you state vegetarian diet can reduce ... via these mechanisms. However, to the right and left of the figure, you talk about non-vegetarian diet. Either change the sentence in the middle or avoid writing about meat-based diet. 

2. The arrows differ in size and length in the figure. If they do not imply something, have them with the same size and length. 

Author Response

  1. Agreed. Made the necessary change in the manuscript.
  2. Agreed. Made the necessary change in the manuscript.

Reviewer 2 Report

The review titled “Plant Based Diet And Its Effect On Cardiovascular Disease” is well written and discussed they established facts, that dietary interventions are used very less than pharmaceutical and procedural interventions in the management of cardiovascular disease. In this review they highlight the role of a healthy plant-based diet in improving cardiovascular outcomes.

Major Comments –

1)     In the introduction, please elaborate on Atherosclerosis and obesity related disorder.

2)     2.1.2. Population- What is the exclusion and inclusion criteria for patient selection in the section 2.1.2?

Author Response

Thank you for reviewing the article

  1. We wanted to focus on Cardiovascular disease and plant based diet. Elaborating on obesity related disorders may shift focus of the readers away from the Cardiovascular aspect. In addition obesity related disorder is a quite sizeable topic and elaborating on it will add a lot of words, and likely exceed word limit.
  2. Agreed. Added to the manuscript.

Reviewer 3 Report

Hi dear

This article " Plant Based Diet and Its Effect on Cardiovascular Disease” was revised and has a novelty and I recommend it for publication after consideration of the following comments.

·       Please include the detail of results in the abstract.

·       Line 37-39: Why only with optimize blood pressure, glycemic and lipid control you suggest for reducing cardiovascular disease. Because from the perspective of science and food industry, bioactive compounds such as antioxidants and natural pigments such as anthocyanin etc. are very effective. Please point to them.

·       Line 50-53:  Line 50-53 is a word for word repetition with the abstract of the article, please improve it.

·       Line 133-135: Please, for the sake of discussion, you can explain why this is the case according to your studies. Otherwise, your article is just stating statistics and is unprintable from the point of view of Journal Foods.

·       Line 148: EPIC-Oxford study? Please avoid of abbreviation in the first time of referring.

·       Line 205: as the same above mention statement.

·       Line 212: Aren't geographic-continental interventions and of course racial factors affecting plant-based diets? How do you explain this?

·       Please, in a paragraph, address the effective research of bioactive plant compounds on cardiovascular health in order to submit a more acceptable article for publication in Journal Foods.

·       Fig 1: Please include the plant’s bioactive e.g., anthocyanin and phenolic compounds etc. in cardiovascular disease preventions.

·       Discussion text must grammar improve and in some cases it is very weak and maybe there is no discussion at all.

·       Conclusions: the final conclusion will be provided.

Author Response

All changes recommended by reviewer were made except:

Point 1. Line 37-39 (hence also not included in Fig 1). This is because data pertinent to bioactive compounds such as antioxidants and natural pigments such as anthocyanin were mostly based on in vitro studies. Their effects in vivo are lacking, hence their direct actions on the cardiovascular system are still controversial. Due to lack of strong evidence in this matter, the team of authors think it is best to avoid discussing its effects since our article focuses more on the clinical (in vivo) perspective and not the food industry (in vitro) perspective.

Referece: Behl T, Bungau S, Kumar K, Zengin G, Khan F, Kumar A, Kaur R, Venkatachalam T, Tit DM, Vesa CM, Barsan G, Mosteanu DE. Pleotropic Effects of Polyphenols in Cardiovascular System. Biomed Pharmacother. 2020 Oct;130:110714. doi: 10.1016/j.biopha.2020.110714. Epub 2020 Sep 28. PMID: 34321158.

Reviewer 4 Report

This review article discussed some cohort and clinical studies of the role of a healthy plant-based diet in improving cardiovascular outcome. The comments for revision are listed below.

1. The #2 reference paper has been cited multiple times in the introduction and played an important role in introducing the main topic of this manuscript – diet and cardiovascular disease, however the reference paper made some strong yet counterintuitive statements without conclusive evidence and seemed to be relatively less cited in the field. This reviewer suggests the authors consider rewriting the introduction with more convincing references introducing why diet is important, if not the most important factor (what about genetics, physical activity etc.), in the risk of cardiovascular disease.

2. In the 2.1.1, it is conflicting that the authors first stated that the driving force behind the Americans’ diet choice of vegetarian or vegan is around the public desire to prolong longevity and avoid the ill-effects of common illnesses, and then used the Adventist Health Study (AHS) as an evidence / example. The AHS cohort is special; can’t the driving force involve religious believes? In this example, the authors also did not mention the effect sizes.

3. In the 2.1.2, for age, median age is a better report than a mean age stat, as age may not be normally distributed in this study. The study is based on questionnaire only, and the study cohort has more elders and women, and thus to some extent, cannot represent a general population, i.e., the study design can be subjected to selection biased, e.g. in a general population, there can’t be more than half are vegetarian-related.

4. In the 2.1.3, the word “specialty diets” lacks a definition, and I don’t see how this conclusion that the “specialty diets had a lower all-cause mortality rate when compared to a standard non vegetarian diet” is derived, given the vegan and semi-vegetarians are both insignificant comparing to non-vegetarians, and the other two groups also just reaches the statistical significance level, not robust when considering multiple testing.

5. In the 2.1.3, in the survival model, are there any confounding factors considered?

6. In the 2.1.3, it is strange why the authors use a logistic regression model rather than a cox model, given the study is a prospective longitudinal study, one should leverage the advantage of years of follow-up.

7. In the 2.1.4, the authors discussed the differences in the vegetarian diet effect between men and women, while did not mention this part of the results in the 2.1.3.  The Discussion section should be based on results included in the Results section.

8.In the 2.1.5, there should be more limitations. Please refer to the comments made above. Overall, by this study alone, it is hard to make the conclusion that vegetarian-related diet is good to reduce all-cause and cardiovascular mortality, especially given the results are statistically non-significant.

9. In the EPIC-Oxford study, there is no demographic characteristics of the study cohort. This should be either included in text or as a table.

10. cholesterol level can be HDL (good) or LDL (bad).

11. In the 2.2.3, “Kinjo et al found…” should be opposite in meaning? And in “This study, in combination with the EPIC and Sun et al study…” also should be opposite in meaning?

12. If there is a meta-analysis, why do the authors illustrate in detail about one study (EPIC-Oxford) of the meta-analysed study rather than review the meta-analysis, or all studies included in the meta-analysis?

13. In the 2.3.4, comparing to previous studies, it is changed to Conclusion, and there is no discussion? The structure for each study should be consistent. Why the weight loss in the second 6 month is much less than the first 6 month?

14. In the 2.4.3 and 2.4.4, only p-values were reported. Effect sizes and their corresponding Cis should also be reported.

15. In the EVADE trial, it is important to discuss that although a vegan diet may better reduce the inflammation than the AHA recommended diet, but a vegan diet did not provide better glycemic control, weight loss or improvement in the lipid profile compared with an AHA recommended diet. It is dangerous to promote vegan diet without a critical assessment of this diet to overall health.

Author Response

All changes were made (in red font) except

Point No. 3 since the paper only disclosed/mentioned mean age, no median ages were available.

Round 2

Reviewer 1 Report

Dear authors,

Hi,

Thanks for implementing the suggested changes. 

Bets regards.

Reviewer 2 Report

The authors have answered to the comments and it can be published in present form. 

Reviewer 3 Report

No further suggestions.

Reviewer 4 Report

All comments are solved. Thanks!